# Recent Advances in Chitin Biosynthesis Associated with the Morphology and Secondary Metabolite Synthesis of Filamentous Fungi in Submerged Fermentation

**DOI:** 10.3390/jof9020205

**Published:** 2023-02-04

**Authors:** Zihan Gong, Song Zhang, Jun Liu

**Affiliations:** Hunan Key Laboratory of Grain-Oil Deep Process and Quality Control, National Engineering Research Center of Rice and Byproduct Deep Processing, Central South University of Forestry and Technology, Changsha 410004, China

**Keywords:** chitin, mycelia morphology, filamentous fungi, submerged fermentation

## Abstract

Metabolites produced by filamentous fungi are used extensively in the food and drug industries. With the development of the morphological engineering of filamentous fungi, numerous biotechnologies have been applied to alter the morphology of fungal mycelia and enhance the yields and productivity of target metabolites during submerged fermentation. Disruption of chitin biosynthesis can modify the cell growth and mycelial morphology of filamentous fungi and regulate the biosynthesis of metabolites during submerged fermentation. In this review, we present a comprehensive coverage of the categories and structures of the enzyme chitin synthase, chitin biosynthetic pathways, and the association between chitin biosynthesis and cell growth and metabolism in filamentous fungi. Through this review, we hope to increase awareness of the metabolic engineering of filamentous fungal morphology, provide insights into the molecular mechanisms of morphological control via chitin biosynthesis, and describe strategies for the application of morphological engineering to enhance the production of target metabolites in filamentous fungi during submerged fermentation.

## 1. The Relationship between Fungal Morphology and Production Performance

Microbial secondary metabolites (MSMs), such as antibiotics, organic acids, hormones, and pigments, typically synthesized during the late growth phase of the producing microorganisms, are generally inessential for their growth [1]. In the year 2000, the market for antimicrobial secondary metabolites was 55 billion dollars [2], and that for antibiotics had increased to 66 billion dollars by 2007 [3]. In addition, Evaluate Pharma predicted that the biopharmaceutical (including hormones, enzymes, and vaccines) market would grow from 202 billion dollars in 2016 to 326 billion dollars in 2022. MSMs, used extensively in the food and pharmaceutical industries, make valuable contributions to health, nutrition, and the social economy. Among the microbial producers of these secondary metabolites, filamentous fungi, distinguished by a mycelium composed of septate hyphae or branching filaments, are the most frequently employed metabolically versatile cell factories in the biotechnology industry, utilized as sources of a diverse range of compounds, including industrial enzymes, penicillin, and citric acid [4,5]. In a previous review of the morphology of industrial fungi, Quintanilla et al. described three main advantages of the widespread application of filamentous fungi: (1) exceptional secretion of proteins or enzymes [6], (2) specialized post-transcriptional modification machinery facilitating glycosylation and correct protein folding [7], and (3) a diverse range of safe species, approved by the regulatory authorities and categorized as generally recognized as safe (GRAS) [4]. Compared with conventional solid-state fermentation, submerged fermentation has notable advantages, including simple parameter control processing, large volume processing, reduced fermentation time, low labor intensity, and ready scale-up, and this type of cultivation is becoming the primary industrial biotechnology used in the production of target fungal metabolites [8,9].

However, there are certain disadvantages associated with the submerged fermentation cultivation of filamentous fungi, such as the high viscosity of the fermentation medium determined by fungal morphology, considerably affecting the transfer of mass, heat, and momentum, and consequently, the yield and productivity of target products within bioreactors [5,10,11]. During submerged cultivation in bioreactors, filamentous fungi grow in one of three morphological forms, namely, dispersed mycelia, clumped aggregates, or mycelial pellets (including rough and smooth pellets) [12,13,14,15]. In addition, during submerged fermentation of filamentous fungi in conical flasks, an irregular mycelial block-type morphology is observed, as seen among species in the genus *Monascus*. Fungal morphology can be markedly and intricately determined by environmental conditions and inherent molecular or genetic biology. Indeed, the occurrence of different morphological forms of fungi when cultivated under submerged fermentation conditions, is one of the several engineering issues that require appropriate resolution. In this regard, Veiter et al. outlined the interlinkages between productivity, performance, and morphology among a range of different fungal species, particularly highlighting the relationship between fungal pellet morphology and metabolite productivity [16]. Figure 1 represents the mechanisms underlying the correlation between environmental conditions or genes, and fungal morphology and metabolite productivity during submerged fermentation. The generation of macroscopic clumps and spherical pellets is associated with a number of undesirable effects, including the limitation of diffusive mass transfer and reduced nutrient and oxygen levels, particularly in the centers of mycelial pellets and clumps. Furthermore, a dispersed mycelial morphology is often associated with an increase in the viscosity of the fermentation medium, thereby reducing mixing efficiency, hampering stirring, and limiting oxygen transport due to the strong non-Newtonian rheological properties of the fermentation medium [17]. Consequently, it is highly desirable to determine the optimal morphology of filamentous fungi during the submerged fermentation process, to enhance the yields of target products in the food and pharmaceutical industries.

Often, it is not clear which morphological form is the most suitable for the efficient production of a given metabolite or enzyme. Dense mycelial pellets, within an appropriate range of diameters, are a favorable morphology for the biosynthesis of citric acid by *Aspergillus niger* [13,18,19], whereas, in *Aspergillus oryzae*, mycelium comprising numerous hyphal tips is more conducive to the production of enzymes, such as α-amylase and lipase [20,21,22]. Furthermore, the dispersed mycelium of *Penicillium chrysogenum* is generally regarded as a prerequisite to ensure the high productivity of penicillin in submerged fermentation [23]. Typically, however, the relationship between fungal morphology and production performance is complex, and different morphological forms have certain associated advantages. In 2001, Mclntyre et al. described the analytical tools employed in the study of hyphal morphology, the physiological aspects of morphological development, and molecular aspects of morphological control of *Aspergillus* [10]; they were the first to propose the term metabolic engineering of the morphology of filamentous fungi (MEMFF), a concept that covers the relationships between cell growth, phenotypes, and target products [10]. Since then, numerous regulatory genes, including the global regulator *Lae*A [24,25], transcription factors *Brl*A and *Wet*A [26,27], CreA [28], and the Zn(II)2Cys6 family [29], have been identified and applied in MEMFF to enhance the production of target metabolites.

Chitin, a major structural component of the fungal cell wall, synthesized via the activity of the enzyme chitin synthase (*chs*), has become a high-profile target for investigating the effect on morphology, yield, and productivity in submerged fermentation. In this review, we summarize the classification of *chs* enzymes, the roles played by *chs* in the cell development and mycelial morphology of filamentous fungi, and the application of MEMFF. We provide a basis for understanding the inter-relationships between fungal differentiation, morphology, and productivity regulated by *chs* genes.

## 2. Chitin and Chitin Biosynthesis in Filamentous Fungi

### 2.1. Structure and Function of Chitin

After cellulose, chitin is the second most abundant natural polysaccharide, occurring widely in the exoskeletons of insects, crustaceans, and mollusks; it is also an important structural polysaccharide in fungal cell walls [30,31,32]. Notably, the chitin content in the fungal cell wall differs according to the morphological phase, accounting for only 1–2% of yeast cell wall dry weight [33,34], but reaching up to 10–20% of the cell wall dry weight of filamentous fungi (*Aspergillus*) [35]. Moreover, the content of chitin in the hyphal walls of *Candida albicans* is three times higher than that of other yeasts [36], whereas, in *Paracoccidioides brasiliensis* and *Blastomyces dermatitidis*, it is 25–30% higher than that in the yeast phase [37]. Chitin is a linear copolymer of *N*-acetyl-d-glucosamine (GlcNAc) and d-glucosamine units, linked by a *β*-(1–4) glycosidic bond, although predominantly comprising GlcNAc units [31]. Chitin chains of more than 100 and 190 GlcNAc monomers in length have been reported in cell walls and bud scars, respectively [33,38]. In addition, crystalline structural determinations have revealed that chitin can exist in three different forms, namely, α-, β-, and γ-chitin, representing antiparallel, parallel, and alternated arrangements of polymer chains, respectively [39,40]. In fungi, α-chitin is the major structural form [41], and γ-chitin is mainly found in the beetle family *Lucanidae* [42].

Chitosan is an important chitin derivative, generated by removing the acetyl group of chitin, either via treatment with concentrated alkali or the activity of chitin deacetylases. Chitin and its derivatives (chitosan and glucosamine series) have important applications in medicine and in the chemical industry, and as functional foods. Chitin and chitosan are considered advantageous biocompatible materials that can be used to augment or replace any tissue, organ, or function of the body [32,43]. Moreover, owing to their notable biological activities, including antibacterial, antifungal, antitumor, immunoregulatory, antioxidant, and anti-inflammatory properties, chitosan oligosaccharides have gained widespread application in the treatment and prevention of multiple life-threatening diseases and disorders, including cancer, heart disease, diabetes mellitus, and serious infections [44]. In addition, chitin and chitosan have a high absorptive capacity for wastewater pollutants, and thus have application potential in industrial wastewater treatment [45]. In filamentous fungi, chitin molecules form intrachain hydrogen bonds, facilitating assembly into fibrous microfibrils that form a basket-like scaffold surrounding cells. These fibrous microfibrils are characterized by considerable tensile strength, thereby maintaining cell wall integrity. As depicted in Figure 2, the cell wall comprises a twin-layer structure, the innermost layer of which is a relatively conserved structural skeletal layer (crosslinked chitin-glucan inner layer) comprising chitin and *β*-(1,3)-branched glucan, whereas the heterogeneous outer layer consists of other polysaccharides and glycoproteins [46,47]. The *β*-(1,3):*β*-(1,6)-branched glucan of the cell wall is bound to proteins or other polysaccharides, the composition of which may vary according to the fungal species, although it generally comprises highly mannosylated glycoproteins and mannoproteins. Chitin plays multiple roles in fungal species, including the maintenance of cell structural integrity, regulation of epithelial adhesion, the linkage between the cell wall and capsule, and antifungal resistance [46,47,48]. Accordingly, chitin is a key factor in maintaining normal cell growth and metabolism in filamentous fungi.

### 2.2. The Chitin Biosynthetic Pathway

In fungi, chitin is synthesized via a highly complex biosynthetic pathway involving a multifarious series of biochemical and physiological processes [49]. As substrates, glucose or one of its storage compounds (glycogen or trehalose) undergoes bioconversion to a polymer of the amino sugar GlcNAc via a series of enzymatic reactions divided into three sets of sub-reactions [49]. In the first set of sub-reactions (Figure 3), the biosynthesis of GlcNAc-1P proceeds via three steps: the substrate, fructose-6-phosphate (fructose-6P), generated from glycolysis (or the Embden–Meyerhof–Parnas (EMP) pathway), and trehalose are mobilized by hydrolysis to glucose catalyzed by trehalase [EC:3.2.1.28], and glycogen is converted to glucose-1-P by glycogen phosphorylase [EC:2.4.1.1]. During this stage, glucokinase [EC:2.7.1.2] or hexokinase [EC:2.7.1.1], and glutamine-fructose 6-phosphate transaminase (isomerizing) [EC:2.6.1.16] are the rate-limiting enzymes. In addition, glucose-6P can be used for the biosynthesis of *β*-(1,3) glucan via three reactions catalyzed by the enzymes phosphoglucomutase [EC:5.4.2.2], UTP-glucose-1-phosphate uridylyltransferase [EC:2.7.7.9], and 1,3-*β*-glucan synthase [EC:2.4.1.34]. In the second set of sub-reactions, GlcNAc-1P is catalyzed to generate the activated molecule amino sugar UDP-*N*-acetylglucosamine (UDP-GlcNAc) via the action of UDP-*N*-acetylglucosamine pyrophosphorylase [EC:2.7.7.23], an essential enzyme for chitin synthesis. In the final set of sub-reactions, the enzyme *chs* [EC:2.4.1.16] catalyzes a polymerization reaction to synthesize chitin using the activated UDP-GlcNAc as a sugar donor. The first two sets of sub-reactions occur within the cell cytoplasm, while the third reaction occurs in the chitosome, located within the plasma membrane of cells in the hyphal tips and cell cross-walls of filamentous fungi [50]. In the chitin biosynthetic pathway, glutamine-fructose 6-phosphate transaminase [EC:2.6.1.16], UDP-*N*-acetylglucosamine pyrophosphorylase [EC:2.7.7.23], and *chs* [EC:2.4.1.16] serve as the rate-limiting enzymes that dictate the rate at which chitin is synthesized, and are highly regulated in cells. Among these enzymes, *chs* catalyzes the final reaction, which is specifically and directly associated with the biosynthesis of chitin, and accordingly, is acknowledged to be the key enzyme in chitin biosynthesis. As described in the Introduction section, *chs* plays a vital role in cell development and the mycelial morphology of filamentous fungi, thereby having a prominent role in the application of MEMFF. Hence, we specifically focused on the chitin synthase in different fungal species and the involvement of *chs*.

### 2.3. Classification of Chitin Synthase

Based on amino acid sequence homology, the *chs* enzyme family can be grouped into seven classes (I to VII), with different fungal species expressing varying numbers of *chs* genes [51]. In 2019, researchers summarized three *chs* genes (classes I–III) in *Saccharomyces cerevisiae*, four in *Candida*, and six to ten in filamentous fungi [52]. Among the seven classes, *CHS* III, V, VI, and VII are found exclusively in filamentous fungi [53]. In the same year, in their review of fungal chitin synthesis and degradation, Yang and Zhang described the members of the seven classes of chitin synthase in different fungi [47]. However, since then, details of *chs* genes and their classification have not been furthered. In this review, we further summarize and update the *chs* gene classes and number of *chs* genes in eukaryon, based on data obtained from the Kyoto Encyclopedia of Genes and Genomes (https://www.kegg.jp/kegg/genes.html) and National Center for Biotechnology Information (https://www.ncbi.nlm.nih.gov/) databases, with an emphasis on those expressed in filamentous fungi. As shown in Table 1, we found differences in the number of *chs* genes in *Saccharomycotina*, although these are generally grouped into three classes (*CHS* I–III). For instance, there are six (as opposed to the originally reported three), five, and four *chs* genes in *S. cerevisiae* S288c, *Candida orthopsilosis* Co 90-125, and *Candida tropicalis* MYA-3404, respectively, whereas eight *chs* genes have been identified in *Sugiyamaella lignohabitans* CBS 10342. Except for those in goldfish (seven *chs* genes), there are generally few *chs* genes in animals, including Eutheria, Amphibia, and Euteleostomi. Notably, the number and classes of *chs* genes in fungal genera are distinctly higher than those in the species of Saccharomycotina and animals. For example, among species of Pezizomycotina, such as *Aspergillus fumigatus*, *Neurospora crassa*, *Cordyceps militaris*, and *Purpureocillium lilacinum*, *chs* genes are generally grouped into seven classes, with seven to nine genes in each. Moreover, certain hypothetical proteins are identified as *chs* enzymes in *Fusarium graminearum* and *Pestalotiopsis fici*, thereby indicating the potential occurrence of up to 10 types of *chs*. Strains of filamentous fungi in the genus *Monascus*, an important industrialized fermentative microorganism, are noted for their production of MSMs, including *Monascus* pigments and monacolin K. In our laboratory, we have sequenced the whole genomes of *M. purpureus* LQ-6 (accession number of PRJNA503091) and its mutant strain M183 (accession number of JAACNI000000000) based on the combined application of single-molecule real-time DNA sequencing and next-generation sequencing. Accordingly, we identified eight genes encoding *chs* enzymes, the classification of which appears to be complex. In addition, nine genes encoding *chs* enzymes (including three hypothetical proteins) have been identified in the genome of *M. purpureus* HQ1 (accession number of VIFY00000000), mainly classified as *CHS* I, II, III, V, and VII. The larger number of *chs* genes in filamentous fungi compared to Saccharomycotina reflects the greater complexity of hyphal development and polarized growth, as well as a higher cell wall chitin content.

With the increasing accumulation of genomic sequence data for fungi in recent years, the number of identified *chs* genes in different species has reached approximately 200. However, most of these genes have yet to be fully characterized. Generally, *chs* enzymes are grouped into two divisions, division I (containing *CHS* I–III) and division II (containing *CHS* IV–VII) [54]. Among the members of the *chs* enzyme family (*CHS* I–VII), 6–10 *chs* genes identified in different fungi species encode proteins with discernable structural differences. As shown in Figure 4, there are obvious differences in the tertiary structures that distinguish the different classes of *chs* proteins. All *chs* members have multiple transmembrane domains (TMD); however, *CHS* IV–VII enzymes typically contain a cytochrome *b*5-like heme/steroid-binding domain (Cyt-*b*5), which is not found in classes I to III. Furthermore, *CHS* V and *CHS* VI proteins both have an *N*-terminal myosin motor domain (MMD) and a C-terminal chitin synthase domain (CSD) [55]. Although the structures of *CHS* V and *CHS* VI proteins are highly similar and difficult to differentiate, the MMD of *CHS* V proteins contains conserved ATP-binding motifs (ABM, including p-Loop, Switch I, and Switch II) absent in class VI chitin synthases [56]. In addition, *CHS* I–III proteins are characterized by hydrophobic C-terminal and hydrophilic N-terminal regions containing a catalytic domain. In our laboratory, the *chs* protein-encoding gene Monascus_05162, detected in the *M. purpureus* LQ-6 genome, was identified as a *CHS* VI class enzyme based on the tertiary structure of the protein and conserved domain analysis [57].

## 3. The Regulatory Effect of Chitin Biosynthesis on the Cell Growth and Morphology of Filamentous Fungi

In recent years, numerous studies have focused on the regulatory effects of chitin biosynthesis on the cell growth and morphology of filamentous fungi. Given its key role in chitin biosynthesis, *chs* has been preferentially selected to investigate its contribution to chitin biosynthesis and cell growth. A decade ago, Kong et al. reported that individual *chs* genes play diverse roles in the hyphal growth and conidiogenesis of the fungus *Magnaporthe oryzae* [58]. Members of the *chs* family have different functional effects with respect to the cell growth, stress tolerances, and cell wall integrity of *Metarhizium acridum*; of these, *CHS* III, V, and VII regulate the surface properties of conidia and hyphal bodies [52]. Kim et al. observed that *CHS* V and *CHS* VII gene knockout in *Gibberella zeae* was associated with the development of balloon-shaped hyphae and weak cell wall rigidity, with the mutants being unable to produce perithecia, thereby inducing disease symptoms in barley heads [59]. Larson et al. found that single or double deletion of *CHS* V and *CHS* VII genes in *Fusarium verticillioides* contributed to the poor growth of mutant strains, and reductions in the diameter and aerial mycelia of colonies cultivated on potato dextrose agar medium [54]. Moreover, when cultured in potato dextrose broth, ballooning of cell walls was observed in the hyphae of all three mutants. However, although the *CHS* V gene in *S. cerevisiae* is not essential for cell growth, it does play an important role in the mating of this yeast [60]. In addition, Amnuaykanjanasin and Epstein indicated that *chs* A (*CHS* V) is essential for conidial wall strength and contributes to the strength of hyphal tips [61]. Collectively, these findings indicate that *CHS* V and *CHS* VII genes are mainly required for normal hyphal growth, perithecia formation, and pathogenicity of filamentous fungi. Moreover, in our previous study, we observed that the deletion of the *CHS* VI gene in the filamentous fungus *M. purpureus* induced the development of balloon-tip-like structures in the hyphae, reduced hyphal branching efficiency, promoted hyphal elongation, and altered mycelial pellet formation during submerged fermentation [57]. Although *CHS* VI knockout reduced the maximum biomass obtained in submerged fermentation cultures by 19.63%, it did not alter the rate at which the colony diameter of *M. purpureus* grew [57]. In contrast, Cui et al. found that the radial growth of *Botrytis cinerea* colonies substantially reduced as a consequence of disrupting the *CHS* VI gene, and speculated that this gene is necessary for appropriate hyphal growth [55]. Similarly, Fajardo-Somera et al. reported that *CHS* VI primarily plays a role in hyphal extension and ascospore formation, based on a comparison of the functional importance of different *CHS* genes with respect to the cell growth and development of *N. crassa* [62]. Besides, it has been reported that inactivation of *chs* 1 (*CHS* I) in *N. crassa* resulted in extensive hyphal swelling and other hyphal abnormalities in liquid medium [63], but no significant changes after disruption of *chs* 2 (*CHS* II, a non-essential chitin synthase) [64]. In addition, the *chs* Z (class VI) gene of *A. oryzae* is specifically involved in hyphal extension and cell wall formation (filamentous cell morphogenesis) [65]. The findings of these studies indicate that *CHS* VI genes primarily play roles in hyphal elongation and determining mycelial morphology, although their contribution in determining the cell growth of filamentous fungi shows interspecific variation.

In this review, we highlight the fact that different *chs* genes have differential regulatory effects on the hyphal growth and morphology of filamentous fungi. For example, *chs* A (*CHS* I) and *chs* C (*CHS* III) genes are inessential for the normal cell growth of *A. fumigatus* and are not implicated in the regulation of colony diameter or growth rate [66]. However, deletion of *chs* B (class III in *Aspergillus nidulans*, class II in *A. fumigatus*) causes severe growth defects, thereby indicating that this gene plays an essential role in determining hyphal tip growth [54,67]. In a recent study on *Verticillium dahlia*, Qin et al. examined the effects of knocking out eight *chs* genes (*chs* 1–8) on cell growth and virulence, and accordingly established the respective differential requirements for these eight genes [68]. Similarly, Hiroyuki Horiuchi cloned six *chs* genes from *A. nidulans* and investigated their effects on cell growth; they found that although the genes play essential roles in growth and morphogenesis, they differ in terms of specific functions [67].

Numerous researchers have focused on the effects of *chs* genes on the cell development and mycelial morphology of industrial filamentous fungi during submerged fermentation. For example, in their investigation of the regulatory effects of chitin synthase on mycelial morphology during submerged fermentation, Sun et al. used an RNAi construct to silence the *chs* C gene in *A. niger* and found that this strategy resulted in the shortening of hyphal length, reduction in the proportion of dispersed mycelia, and increase in the compactness of mycelial pellets [69]. In subsequent studies, the same group established that the function of the *chs* C gene is tightly interrelated with the functioning of the chitin synthase activator (*chs*3) gene of *A. niger*, and that the desired mycelial morphology for enhancing citric acid production in submerged fermentation cultures can be obtained by knockdown of the *chs*3 gene [70]. In addition, disruption of the *chs* B gene in *A. oryzae* resulted in a significant reduction in the formation of mycelial clumps and corresponding increases in the number of freely dispersed hyphae and frequency of branching [71]. Conversely, deletion of the *csm* A (class V) gene resulted in a reduction in branch number in the apical compartment, along with an increase in the average diameter of hyphae [71]. Moreover, Liu et al. established that the *chs* 4 (class III) gene is essential for the hyphal growth and conidial development of *P. chrysogenum*, and observed a marked reduction in the diameters of mutant colonies following the knockdown of this gene using RNA silencing [72]. Furthermore, these researchers also established that by disrupting *chs* 4 gene expression, they could regulate the agglomeration of hyphal elements and diameters of *P. chrysogenum* mycelial pellets during fermentation [23]. More recently, our research group has been focusing on the morphological changes in *M. purpureus* during submerged fermentation, and we found that disruption of the biosynthetic pathways of both ergosterol (by deleting the *egr* 4A and *erg* 4B genes) and chitin (by knocking out a *CHS* VI gene) promoted changes in the morphology of the mycelial pellets [8,57].

To summarize, in the relevant studies conducted on chitin synthase to date, researchers have primarily investigated the relationships between cell wall chitin content and pathogenesis or tolerance, based on the deletion of *chs* genes [54,58,66,73], and evaluated the regulatory effect of different members of the *chs* family on mycelial morphology in submerged fermentation, with the aim of enhancing the yields of target products [4,57,69,71,72]. Based on the findings of these studies, chitin biosynthesis is closely associated with the cell growth and morphology of filamentous fungi, and different members of the *chs* family play distinct functional roles in this regard.

In addition to the direct effects of *chs* genes, the regulation of chitin biosynthesis is also determined to differing extents by other factors, including transcription factors, mitogen-activated protein kinase (MAPK), and calcium (Ca^2+^). In 2020, Norio Takeshita reported that the oscillation of Ca^2+^ levels can contribute to the control of chitin biosynthesis associated with the stepwise extension of hyphal tips [74], which is consistent with the findings of earlier studies indicating that intracellular Ca^2+^ levels are closely associated with the polarized growth of filamentous fungi [75,76]. In addition, the transcription factors *Crz*A and *Rlm*A, and MPKA, regulate the expression of *chs* genes, thereby altering the growth phenotypes of *A. fumigatus* [77]. Besides, He et al. deleted eki gene encoding ethanolamine kinase in *Trichoderma reesei*, causing the overexpression of chitin synthase genes and increased chitin content; the results also indicate that ethanolamine kinase has a significant effect on cell growth and development in filamentous fungi *T. reesei* [78].

Many signaling pathways, including the high osmolarity glycerol, protein kinase C (PKC)-MAPK, and Ca^2+^/calcineurin pathways, are implicated in the regulation of chitin synthase [79], and chitin biosynthesis is mediated by the regulatory factors *Aba*A, *Brl*A, and *Med*A, which control the levels of *chs*C and *chs*A transcription during conidiophore development [80,81]. Moreover, deletion of the *kex*B gene in *A. niger* resulted in hyper-branching and thickening of cell walls, and upregulated chitin metabolism during submerged fermentation [82]. Based on the evidence accumulated to date, chitin biosynthesis and cell growth are intricate processes regulated via multiple pathways and diverse factors during cell development. Undoubtedly, however, much remains to be discovered; thus, the respective relationships between chitin biosynthesis (*chs* genes) and mycelial morphologies of diverse filamentous fungi, as well as the regulation of chitin biosynthesis via other pathways, should be further studied.

## 4. The Application of Morphological Engineering of Industrial Filamentous Fungi

MEMFF can facilitate substantial improvements in the yield and productivity of target metabolites, thereby contributing to the development of industrial-scale production. As shown in Table 2, a diverse range of technologies have been applied in developing MEMFF for enhancing the target production or yield. In this regard, a series of papers have indicated that extractive fermentation via the addition of surfactants can alter the mycelial morphology of *Monascus* species, thereby significantly enhancing the production of *Monascus* pigments [83,84,85,86]. Furthermore, based on the integrated control of operational parameters (agitation speed and aeration rate) and overexpression of tyrosine-protein phosphatase, Chen et al. fine-tuned the morphology of *A. oryzae* to obtain a more compact pellet structure, and larger pellet number, than the original mycelial morphology in submerged fermentation, thereby increasing the production of l-malate to 142.5 g/L in a 30-L bioreactor [87]. Moreover, the *pka*C gene has multiple regulatory effects associated with hyphal growth, and the overexpression of this gene modified the mycelial pellet morphology of *A. niger* during submerged fermentation, thus contributing to a 1.87-fold increase in the concentration of citric acid [19]. More recently, microparticle-enhanced cultivation techniques have been successfully applied in the submerged fermentation cultivation of different fungal genera. For example, the addition of silicate microparticles controlled the morphological development of *A. niger*. The formation of freely dispersed mycelia promoted 4-fold increases in the concentrations of glucoamylase and fructofuranosidase in shaking flask cultures and production of up to 160 U/mL fructofuranosidase in a 3.0-L stirred tank bioreactor [88]. Similarly, precise control of the morphology of *M. purpureus* has been achieved by the addition of 4 g/L 5000-mesh talc to a submerged fermentation broth at 24 h. In this fermentation system, yields of up to 554.2 U/mL *Monascus* yellow pigments were obtained, representing an approximate 113.15% increase compared with the control group [89].

The following is a summary of the application of MEMFF to enhance the production of fungal target metabolites based on the modification of chitin biosynthesis. Liu et al. reported that in *P. chrysogenum*, pellets and highly branched hyphae may be the most suitable mycelial morphologies for penicillin production, based on the modified expression of the *chs* 4 (class III) gene, following which, yields of penicillin increased by 41% [23,72]. More than 20 years ago, the group headed by Mhairi Mclntyre published a comprehensive review of the MEMFF of *Aspergillus*, in which they described in detail the phenotypic effect of *chs* genes [10]. Subsequently, this research team developed an MEMFF system for *A. oryzae* based on disruption of the *chs*B and *csm*A (class V) genes, which despite contributing to an increase in hyphal branching, did not enhance α-amylase production [72]. However, deletion of the *chs*C gene in *A. niger* significantly modified the mycelial morphology in submerged fermentation, contributing to a 42.6% increase in citric acid production compared with that produced by the wild-type strain [69]. Furthermore, the mycelial morphology of *A. niger* in submerged cultures can be optimized by silencing the chitin synthase activator (*chs*3) gene, resulting in a 39.25% increase in the production of citric acid [70]. Notably, our research group has found that the total production of microbial secondary metabolites (*Monascus* pigments and citrinin) by *M. purpureus* was significantly reduced in response to the disruption of the *CHS* VI gene; however, downregulation of the expression levels of different *chs* genes to appropriate levels can substantially enhance metabolite production [57]. Collectively, these findings indicate that regulation of the mycelial morphologies of filamentous fungi is a complex, multifarious phenomenon, involving multiple genes and pathways. To date, the application of MEMFF based on the disruption of *chs* genes has been applied primarily with the aim of enhancing the citric acid and penicillin production of *A. niger* and *P. chrysogenum,* respectively. However, as proposed by Liu et al., it is reasonable to expect that the manipulation of *chs* enzymes in other fungal species would be a simple and low-cost strategy for enhancing the production of target metabolites synthesized by industrial filamentous fungi under submerged fermentation conditions [23].

## 5. Conclusions

Filamentous fungi with certain unique properties have been widely applied in the food and pharmaceutical industries to produce target metabolites based on submerged fermentation cultivation. The yield and productivity of target products synthesized by filamentous fungi under submerged fermentation conditions are closely associated with fungal mycelial morphology, which has a significant influence on the transfer of mass, heat, and momentum. The mycelial morphologies of filamentous fungi are typically diverse and influenced by multiple factors, including environmental conditions, operating parameters, and autologous genes, thereby contributing to the complex relationship between morphology and production. Chitin, a major structural component of the cell wall of filamentous fungi, synthesized by the activity of chitin synthase, has become a high-profile target for investigating the factors affecting fungal morphology, yield, and productivity in submerged fermentation. In this review, we comprehensively summarize the classification and structures of members of the chs family, and describe the biosynthetic pathways of chitin and the associations between chitin biosynthesis, cell growth, and the metabolism of filamentous fungi. We also summarize studies that have focused on the manipulation of fungal chs genes, describing how the disruption of class III and V chs genes can significantly optimize mycelial morphologies during submerged fermentation of species of Aspergillus, thereby enhancing the production of target metabolites. Regulation of the expression levels of other genes and pathways (including the regulatory factors AbaA, BrlA, and MedA, and the PKC-MAPK and Ca^2+^/calcineurin pathways) can influence the biosynthesis of chitin and cell development, a knowledge of which could be usefully applied in MEMFF to enhance the yields of target products. However, although numerous strategies have been devised to facilitate the control or modification of the mycelial morphologies of filamentous fungi during submerged fermentation, at present, the application of MEMFF based on controlling the expression levels of certain chs genes is not sufficiently comprehensive. Accordingly, further studies are warranted to gain a broader understanding of the regulatory effects of other chs genes on different fungal species and the underlying molecular mechanisms.

## Figures and Tables

**Figure 1 jof-09-00205-f001:**
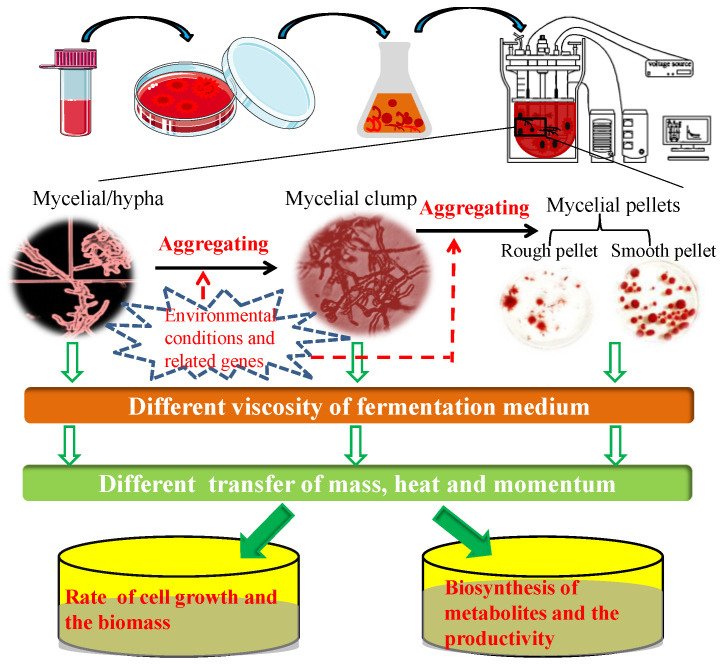
Mechanisms underlying the correlations among environmental conditions or genes, fungal morphology, and the production of metabolites during submerged fermentation.

**Figure 2 jof-09-00205-f002:**
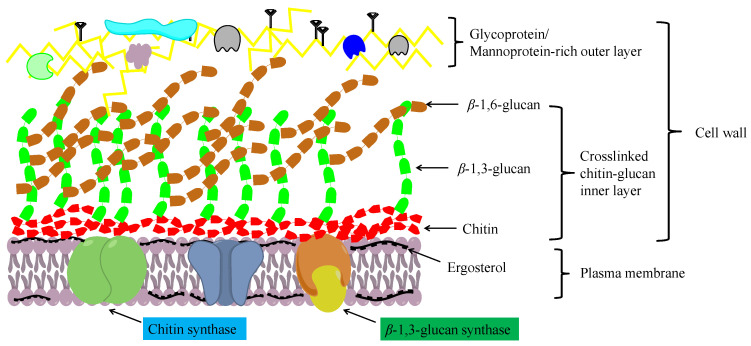
A schematic diagram showing the structure of the fungal cell wall.

**Figure 3 jof-09-00205-f003:**
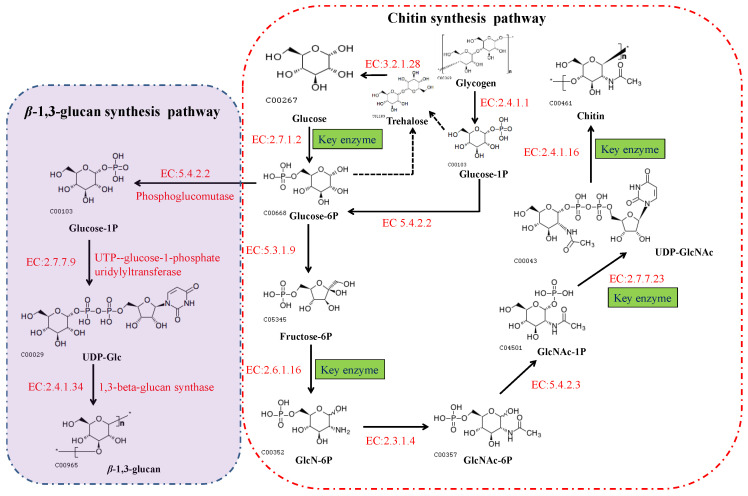
The biosynthetic pathways of chitin and *β*-(1,3) glucan in fungi. These pathways can also be viewed at the Kyoto Encyclopedia of Genes and Genomes website (https://www.kegg.jp/pathway/map00520, accessed on 11 September 2021).

**Figure 4 jof-09-00205-f004:**
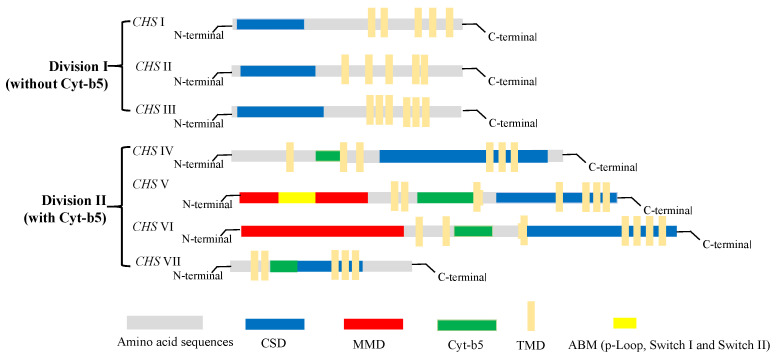
The structure and classification of members of the chitin synthase family.

**Table 1 jof-09-00205-t001:** The members of the chitin synthase family in a section of diverse fungi. (Completely statistical data are shown in Appendix A). *chs*, represents the gene of chitin synthase; *CHS*, represents the class of the members of *chs* family. The genes encoding hypothetical proteins, but mostly like *chs*, are marked in red.

Organism	T-Number	The Members of Chitin Synthase										Number of Genes
*Saccharomyces cerevisiae* S288c	T00005	YBR023C, *chs* 3	YBR038W, *chs* 2	YNL192W, *chs* 1	YLR330W, *chs* 5	YJL099W, *chs* 6	YHR142W, no KO assigned | (RefSeq) *chs*7; *chs* 7p					6
*Lodderomyces elongisporus* NRRLYB-4239	T01116	LELG_05384, *chs* 2	LELG_05013, *chs* 1	LELG_02210, *chs* 2	LELG_00298, *chs* 3	LELG_00300, *chs* 3						5
*Candida tropicalis* MYA-3404	T01115	CAALFM_ C113110CA, *chs* 3	CAALFM_ C300710WA, *chs* 8	CAALFM_ C702770WA, *chs* 1	CAALFM_ CR09020CA, *chs* 2							4
*Candida orthopsilosis* Co 90-125	T02488	CORT_0A01870, *chs* 3	CORT_0D06430, chs 8	CORT_0G01660, *chs* 2	CORT_0H01960, *chs* 1	CORT_0H01970, *chs* 1						5
*Sugiyamaella lignohabitans* CBS 10342	T05270	AWJ20_11, *chs* 6	AWJ20_12, *chs* 3	AWJ20_13, *chs* 3	AWJ20_1163, *chs* 2	AWJ20_1500, *chs* 2	AWJ20_3769, *chs* 1	AWJ20_4861, *chs* 3	AWJ20_4948, *chs* 3			8
*Xenopus laevis* (African clawed frog)	T01010	108717413, *chs* 2	108716131, *chs* 2									2
*Xenopus tropicalis* (tropical clawed frog)	T01011	105947355, *chs* 2-like isoform X1										1
*Carassius auratus* (goldfish)	T07313	113057339 *CHS* 2-like	113061218 *CHS* 1-like	113061224 *CHS* 1-like	113061225 *CHS* 1-like	113061526 *CHS* 1	113061527 *CHS* 1-like	113113123 *CHS* 2-like				7
*Pyricularia oryzae* 70-15	T01027	MGG_09962, *chs* 4	MGG_06064, *chs* D	MGG_09551, *chs* 3	MGG_13013, *chs* 8	MGG_13014, *CHS* V	MGG_01802, *chs*1	MGG_04145, *chs* 2				7
*Fusarium graminearum*	T01038	FGSG_01272, *chs* 4	FGSG_01949, *chs* D	fgr:FGSG_12039, *chs* 6	fgr:FGSG_01964, hypothetical protein	fgr:FGSG_02483, *chs* 2	fgr:FGSG_10116, *chs* 1	fgr:FGSG_10327, *chs* 3	fgr:FGSG_10619, hypothetical protein	fgr:FGSG_03418, *chs* 1	fgr:FGSG_06550, hypothetical protein	10
*Purpureocillium lilacinum*	T05029	VFPFJ_00650, *chs* D	VFPFJ_00666, *chs* 6	VFPFJ_00667, *chs* 6	VFPFJ_03324, *chs* D	VFPFJ_04443, *chs* A	VFPFJ_08553, *chs* G	VFPFJ_08866, *chs* A	VFPFJ_11040, chs			8
*Pestalotiopsis fici* W106-1	T04924	PFICI_01118, *chs* 1	PFICI_01446, *chs* 4	PFICI_04362, hypothetical protein	PFICI_04363, hypothetical protein	PFICI_05017, *chs* D	PFICI_05238, *chs* 2	PFICI_06085, *chs* 3	PFICI_07201, *chs* 1	PFICI_12982, hypothetical protein	PFICI_13513, *chs* 1	10
*Botrytis cinerea* B05.10	T01072	BCIN_01g02520, *CHS* IIIb	BCIN_01g03790, *CHS* IV	BCIN_04g03120, *CHS* IIIa	BCIN_07g01300, *CHS* VII	BCIN_09g01210, *CHS* I	BCIN_12g01380, *CHS* II	BCIN_12g05360, *CHS* VI	BCIN_12g05370, *CHS* V			8
*Aspergillus fumigatus* Af293	T01017	AFUA_4G04180, *chs* B	AFUA_8G05630, *chs* F	AFUA_5G00760, *chs* C	AFUA_2G01870, *chs* A	AFUA_1G12600, *chs* D	AFUA_3G14420, *chs* G	AFUA_2G13430, *chs*	AFUA_2G13440, *chs* E			8
*Aspergillus niger* CBS 513.88	T01030	ANI_1_316024, chs	ANI_1_2332024, *chs*	ANI_1_1542034, *chs* C	ANI_1_684064, *chs* C	ANI_1_1986074, *chs* D	ANI_1_252084, *chs* D	ANI_1_498084, *chs* B	ANI_1_1214104, *chs* C	ANI_1_120124, *chs* A		9
*Aspergillus nidulans* FGSC A4	T01016	AN1555.2, *CHS* V (*chs* D)	AN2523.2, *chs* B	AN4367.2, hypothetical protein	AN4566.2, hypothetical protein	AN6317.2, hypothetical protein	AN6318.2, hypothetical protein	AN7032.2, hypothetical protein				7
*Neurospora crassa*	T01034	NCU09324, *chs* 4	NCU04352, *chs* 5	NCU04350, *chs* 6	NCU05268, *chs* 6;	NCU05239, *chs* A	NCU03611, *chs* 1	NCU04251, *chs* 3				7
*Penicillium digitatum* Pd1	T04849	PDIP_79230, *chs* E	PDIP_62350, hypothetical protein	PDIP_46630, *chs* G	PDIP_26990, *chs* D	PDIP_24450, *chs* G	PDIP_15450, *chs* B	PDIP_07640, *chs* A	PDIP_03360, *chs* F			9
*Coccidioides immitis* RS	T01114	CIMG_05021, *CHS* V	CIMG_05598, *chs* C	CIMG_05647, *chs* G	CIMG_05022, *chs* 5	CIMG_08766, *chs* 4	CIMG_08655, *chs* 2	CIMG_06862, *CHS* VI				8
*Monascus purpureus* HQ1		TQB77221.1, *CHS* V	TQB75461.1, *CHS* III	TQB73913.1, *CHS* I	TQB72986.1, *CHS* VII	TQB70564.1, *CHS* II	TQB69157.1, *CHS* II	TQB73548.1, hypothetical protein	TQB73973.1, hypothetical protein	TQB73547.1, hypothetical protein		9
*Monascus purpureus* LQ-6		monascus_02563, *chs*2	monascus_02508, *chs*3	monascus_05,161 *chs* 4	monascus_05162, *chs* 6	monascus_02870, *chs* activator	monascus_02765, *chs* 5	monascus_02400, *chs* G	monascus_04382, *chs* A			8
*Monascus purpureus* M183		g872, *chs* 2	g920, *chs* F	g3077, *chs*E	g3078, *chs*	g2747, *chs* 3	g5275, *chs* 3	g4739, *chs* B	g5640, *chs* A			8

**Table 2 jof-09-00205-t002:** A selection of the technologies applied in metabolic engineering of the morphology of filamentous fungi for enhancing the production of target metabolites.

Species.	Technology	Mycelial Morphology	Production/Concentration of Target Metabolite	Reference
*M. purpureus*	Control of shakingspeed and pH	Small mycelial pellets with shorter and thickermulti-branched hyphae	The production of yellow *Monascus* pigmentsincreased to 401 U/mL	[84]
*M. purpureus*	Addition of soybean oiland Span-80	Multi-branched hyphae witha number of vesicles	The production of yellow *Monascus* pigmentsincreased by 26.8-fold	[85]
*A. oryzae*	Control of operational parametersand overexpression of tyrosine-proteinphosphatase	More compact pellet structure	The production of l-malate increased to142.5 g/L	[86]
*A. niger*	Overexpression of the *pka*C gene	Modified mycelial pellets	The production of citric acid increased by up to1.87-fold	[19]
*A. niger*	Addition of silicate microparticles	Freely dispersed mycelium	The concentrations of glucoamylase andfructofuranosidase increased by 4-fold	[87]
*M. purpureus*	Addition of 5000-mesh talc	Small mycelial pellets	The yield of *Monascus* yellow pigmentsincreased by 113.15%	[88]
*P. chrysogenum*	Deletion of *chs*4	Pellets and highly branched hyphae	Penicillin production increased by 41%	[23,72]
*A. oryzae*	Deletion of *chs*B	Highly branched hyphae	No effect on α-amylase production	[71]
*A. niger*	Deletion of *chs*C	Compact mycelial pellets	Citric acid production increased by 42.6%	[69]
*A. niger*	Deletion of *chs*3	Higher number of smoother pellets	Citric acid production increased by 39.25%	[70]
*M. purpureus*	Deletion of *chs* VI	Highly rough mycelial pellets	*Monascus* pigments production was reducedby more than 75%	[57]

## Data Availability

The data presented in this study are available upon request from the corresponding author.

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
