# Peer review of "Recent Advances in Chitin Biosynthesis Associated with the Morphology and Secondary Metabolite Synthesis of Filamentous Fungi in Submerged Fermentation"

_jof, 2023, doi:10.3390/jof9020205_

Round 1
Reviewer 1 Report
As a review article about Chitin Biosynthesis associated with the secondary metabolite synthesis of filamentous fungi, it did not cover any information from two important SMS models in the fermentation industry, Trichoderma and Chaetomium spp., is not acceptable, and studies on chitin biosynthesis molecular biology from leading laboratories such as Louise Glass team on Neurospora crassa or Oded Yarden team on Trichoderma should also be reviewed and discussed. Also, the authors need to be very careful about their citations, for example, one of the most important study by Mclntyre et al. 2001 for MEMFF was listed as two references (10 and 89) and was cited as Nielsen et al (line 88) is not acceptable.
Some minor points:
1) Please consider "submerged fermentation" in the title.
2)Please briefly define "filamentous fungi" early in the introduction (before line 36).
3) Please make the information of lines 26-33 directly relevant to fungi, rather than microbial.
4) Please provide ref(s) for lines 44-46, although it is very common in many fungi.
5) Lines 57-58, not correct in biological sense, as morphology is determined by the inherent molecular and genetic biology, not "influenced".
6) Line 71, please explain non-Newtonian rheological properties.
7) Line 88, please use the first author's name for citation!
8) Line 120-121, read strange, and yeasts are fungi!
9) Line 189-190, you cited a 2019 paper, which summarized those information, not "the authors identified those genes"!
10) Line 191, please be consistent with "classes" instead of using "groups".
11) Line 194, furthered for what?
12) Line 218-219, don't understand the English.
13) Table 1, please explain why some texts were highlighted in red using table notes.
14) Line 231, what did you mean by "fully characterized", is it necessary for most of the genes identified by ortholog grouping?
15) Lines 253-254, 303-304, please avoid "numerous", unless you did cited a lot of studies!
16) Lines 291-292, please clarify that was among "different fungi" or in the "same fungus".
17) Lines 365-367, how was "tyrosine-protein phosphatase overexpression" relevant to your story about Chitin synthesis?
Author Response
Response to Reviewer 1 Comments
As a review article about Chitin Biosynthesis associated with the secondary metabolite synthesis of filamentous fungi, it did not cover any information from two important SMS models in the fermentation industry, Trichoderma and Chaetomium spp., is not acceptable, and studies on chitin biosynthesis molecular biology from leading laboratories such as Louise Glass team on Neurospora crassa or Oded Yarden team on Trichoderma should also be reviewed and discussed. Also, the authors need to be very careful about their citations, for example, one of the most important study by Mclntyre et al. 2001 for MEMFF was listed as two references (10 and 89) and was cited as Nielsen et al (line 88) is not acceptable.
Response: Thank you very much for your good questions and advices. Sorry for our carelessness, we have carefully checked the references and revised them. What’s more, Metabolic engineering of the morphology of Trichoderma and Chaetomium by disrupting chitin synthase have not been reported, so we lose sight of them in the original manuscript. As they are important industrial strains, we have added the genes of chitin synthase (seen in Table S1) and the regulation of chitin biosynthesis by some factors from Trichoderma and Chaetomium in the revised manuscript. In addition, we have added more contents from Neurospora crassa in the revised manuscript.
Some minor points:
- Please consider "submerged fermentation" in the title.
Response: Thank you very much for your good advice, we have added these terms in the revised manuscript.
- Please briefly define "filamentous fungi" early in the introduction (before line 36).
Response: Thank you very much for your question, we have defined it as “filamentous fungi distinguished by a mycelium composed of septate hyphae or branching filaments” in the revised manuscript.
- Please make the information of lines 26-33 directly relevant to fungi, rather than microbial.
Response: Thank you very much for your good advice. We wrote the contents of lines 26-33 to summarize the information of secondary metabolites and furtherly introduce that produced by filamentous fungi. Beside, these introduced references used “microbial” but not the specific type of fungi. Thus, it is “microbial”.
- Please provide ref(s) for lines 44-46, although it is very common in many fungi.
Response: Thank you very much for your question, the refs were showed as 8 and 9.
- Lines 57-58, not correct in biological sense, as morphology is determined by the inherent molecular and genetic biology, not "influenced".
Response: Thank you very much for your question, we have revised it to “Fungal morphology can be markedly and intricately determined by….”.
- Line 71, please explain non-Newtonian rheological properties.
Response: Thank you very much for your question. In these contents, we showed that mycelial morphology is often associated with an increase in the viscosity of the fermentation medium, and introduced the ref 17, in which the concept of non-Newtonian rheological properties was used directly.
- Line 88, please use the first author's name for citation!
Response: Thank you very much for your question, we have revised it to “Mclntyre et al…”.
- Line 120-121, read strange, and yeasts are fungi!
Response: Thank you very much for your question, we have revised it to “In fungi, α-chitin is the major structural form [42], and γ-chitin is mainly found in the beetle family Lucanidae [43]”, and changed the ref 43.
- Line 189-190, you cited a 2019 paper, which summarized those information, not "the authors identified those genes"!
Response: Thank you very much for your question, we have revised this sentence to “In 2019, researchers summarized three chs genes…”.
10) Line 191, please be consistent with "classes" instead of using "groups".
Response: Thank you very much for your advice, we have revised it.
11) Line 194, furthered for what?
Response: Thank you very much for your question, the statement was complete, it same as the sentence “However, since then, details of chs genes and their classification have not been furtherly reported”.
12) Line 218-219, don't understand the English.
Response: it means that there are eight chs genes in Monascus purpureus LQ-6 and M183, and the classes of chs family are complex (seen table 1).
13) Table 1, please explain why some texts were highlighted in red using table notes.
Response: Thank you very much for your question. We have added “Red color represents the genes encoding hypothetical proteins and no KO assign, but most likely are chitin synthases.” in the legend of table 1.
14) Line 231, what did you mean by "fully characterized", is it necessary for most of the genes identified by ortholog grouping?
Response: Thank you very much for your question. It means that genes are just identified as the chitin synthase, but still not classified as CHS I~VII. Yes, we have illustrated that the fact that different chs genes have differential regulatory effects on the hyphal growth and morphology of filamentous fungi.
15) Lines 253-254, 303-304, please avoid "numerous", unless you did cited a lot of studies!
Response: Thank you very much for your advice. In fact, line 253-254, 303-304 were the conclusions, and following many studies were expressed and referenced in the corresponding places. Thus, we did no delete or change the term “numerous”.
16) Lines 291-292, please clarify that was among "different fungi" or in the "same fungus".
Response: Thank you very much for your question. From the introduced studies, different chs genes have differential regulatory effects in different and same fungi. Thus, we described it as “we highlight the fact that different chs genes have differential regulatory effects on the hyphal growth and morphology of filamentous fungi.”
17) Lines 365-367, how was "tyrosine-protein phosphatase overexpression" relevant to your story about Chitin synthesis?
Response: Thank you very much for your question. Here, we just showed the application of MEMFF for enhancing target production to furtherly introduce the construction of MEMFF by disruption of chs genes.

Reviewer 2 Report
This is an interesting review concentrating on chitin synthesis and its regulation in fungi. The authors explore the idea that control of fungi morphology, via interference with chitin biosynthesis, contributes to productivity. I recommend it for publication in the present format.
Author Response
Dear reviewer,
Thank you very much for your recognition to our study.
Thank you and best regards.
Yours sincerely,
Jun Liu (Ass. Prof., Dr.)
College of Food Science and Engineering, National Engineering Research Center of Rice and Byproduct Deep Processing, Central South University of Forestry and Technology, Changsha, Hunan 410004, China.
Round 2
Reviewer 1 Report
I am satisfied with the revision and the authors' responses to my comments.